# The skin is a significant but overlooked anatomical reservoir for vector-borne African trypanosomes

**Paul Capewell[1,2,3†], Christelle Cren-Travaillé[4,5†], Francesco Marchesi[6], Pamela Johnston[6], Caroline Clucas[1,2,3], Robert A Benson[2,7,8], Taylor-Anne Gorman[1,2,3,7,8], Estefania Calvo-Alvarez[4,5], Aline Crouzols[4,5], Grégory Jouvion[9], Vincent Jamonneau[10], William Weir[1,2,3], M Lynn Stevenson[6], Kerry O'Neill[1,2,3], Anneli Cooper[1,2,3], Nono-raymond Kuispond Swar[11], Bruno Bucheton[10], Dieudonné Mumba Ngoyi[12], Paul Garside[2,7,8], Brice Rotureau[4,5‡], Annette MacLeod[1,2,3*‡]**

[1]Wellcome Trust Centre for Molecular Parasitology, University of Glasgow, Glasgow, United Kingdom; [2]College of Medical, Veterinary and Life Sciences, University of Glasgow, Glasgow, United Kingdom; [3]Henry Wellcome Building for Comparative Medical Sciences, University of Glasgow, Glasgow, United Kingdom; [4]Trypanosome Transmission Group, Trypanosome Cell Biology Unit, INSERM U1201, Paris, France; [5]Department of Parasites and Insect Vectors, Institut Pasteur, Paris, France; [6]Veterinary Diagnostic Services, Veterinary School, University of Glasgow, Glasgow, United Kingdom; [7]Institute of Infection, Immunology and Inflammation, University of Glasgow, Glasgow, United Kingdom; [8]Glasgow Biomedical Research Centre, University of Glasgow, Glasgow, United Kingdom; [9]Human Histopathology and Animal Models Unit, Institut Pasteur, Paris, France; [10]Institut de Recherche pour le Développement, Unité Mixte de Recherche IRD-CIRAD 177, Campus International de Baillarguet, Montpellier, France; [11]University of Kinshasa, Kinshasa, Democratic Republic of the Congo; [12]Department of Parasitology, National Institute of Biomedical Research, Kinshasa, Democratic Republic of the Congo

*For correspondence: annette.macleod@glasgow.ac.uk

†These authors contributed equally to this work
‡These authors also contributed equally to this work

**Competing interests:** The authors declare that no competing interests exist.

**Abstract** The role of mammalian skin in harbouring and transmitting arthropod-borne protozoan parasites has been overlooked for decades as these pathogens have been regarded primarily as blood-dwelling organisms. Intriguingly, infections with low or undetected blood parasites are common, particularly in the case of Human African Trypanosomiasis caused by *Trypanosoma brucei gambiense*. We hypothesise, therefore, the skin represents an anatomic reservoir of infection. Here we definitively show that substantial quantities of trypanosomes exist within the skin following experimental infection, which can be transmitted to the tsetse vector, even in the absence of detectable parasitaemia. Importantly, we demonstrate the presence of extravascular parasites in human skin biopsies from undiagnosed individuals. The identification of this novel reservoir requires a re-evaluation of current diagnostic methods and control policies. More broadly, our results indicate that transmission is a key evolutionary force driving parasite extravasation that could further result in tissue invasion-dependent pathology.

## Introduction

Understanding the process of parasite transmission is essential for the design of rational control measures to break the disease cycle and requires the identification of all reservoirs of infection. In a number of vector-borne diseases, it is becoming evident that asymptomatic individuals, be they humans or animals, can represent a significant proportion of the infected population and therefore an important reservoir of disease that requires targeting by control measures (*Lindblade et al., 2014*; *Fakhar, 2013*; *Koffi et al., 2006*; *Berthier et al., 2016*). The recent identification in West Africa of asymptomatic individuals with human trypanosomiasis (long-term seropositives) but undetected parasitaemia, raises the question of what role these individuals play in disease transmission (*Koffi et al., 2006*; *Jamonneau et al., 2012*; *Bucheton et al., 2011*; *Kanmogne et al., 1996*). Therapy is currently only directed towards microscopy-positive individuals and thus a proportion of the infected population remain untreated.

There is convincing evidence that seropositive individuals with low or undetected parasitaemia contain transmissible trypanosomes. Xenodiagnosis experiments, in which tsetse flies are fed on microscopy-negative infected humans (*Frezil, 1971*) or, more recently, experimentally-infected pigs (*Wombou Toukam et al., 2011*), have shown that these apparently aparasitaemic hosts contain the parasite since the tsetse flies became infected. It is uncertain where the trypanosomes reside in the host but, given the telmophagus (slash and suck) feeding habit of the tsetse fly, they could be skin-dwelling parasites ingested with the blood meal. Our findings suggest that parasites may be sufficiently abundant in the skin to allow transmission and therefore the skin may represent an anatomical reservoir of infection.

Detection of trypanosomes in the skin is not well documented, although there are descriptions of cutaneous symptoms associated with African trypanosomiasis and distinct 'trypanid' skin lesions (*McGovern et al., 1995*). Imaging data from mouse models of infection suggest that trypanosomes sequester to major organs such as the spleen, liver and brain (*Blum et al., 2008*; *Kennedy, 2004*) and recent evidence has demonstrated trypanosomes in extravascular adipose tissue (*Trindade et al., 2016*). These adipose-associated trypanosomes appear to be a new life-cycle stage with a distinct transcriptional profile and, while tsetse bite-site associated transmission has been suggested (*Caljon et al., 2016*), and a historical study made a passing observation of localised deposition of trypanosomes in the skin matrix (*Goodwin, 1971*), the broader role of skin-dwelling trypanosomes in transmission remains unclear. In this paper we report the investigation of a possible anatomical reservoir in the skin of the mammalian host. We provide conclusive evidence of *T.b. brucei*, (a causative agent of animal trypanosomiasis) and the human-infective trypanosome, *T.b. gambiense*, invading the extravascular tissue of the skin (including but not restricted to the adipose tissue) and undergoing onward transmission despite undetected vascular parasitaemia. We also provide evidence of localisation of trypanosomes within the skin of undiagnosed humans. The presence of a significant transmissible population of *T. brucei* in this anatomical compartment is likely to impact future control and elimination strategies for both animal and human trypanosomiases.

## Results

In order to investigate the potential for extravascular skin invasion by *T. brucei*, BALB/c mice were inoculated via IP injection with the low virulence STIB247 strain of *T. brucei* and skin sections were assessed over a 36-day time-course. The presence and relative quantities of extravascular parasites were evaluated by semi-quantitative scoring of the histological samples (*Figure 1—source data 1*) and compared to blood parasitaemia (*Figure 1—source data 2*). Extravascular parasites were first observed in the skin 12 days post-infection and remained throughout the experiment. Skin parasite numbers fluctuated to a lesser extent than blood parasitaemia and the apparent periodicity in the skin may be due to one particularly high data point on day 24 (*Figure 1*). Parasites were found in the dermis, subcutaneous adipose tissue (*Figure 2*) and in fascia beneath the panniculus carnosus muscle. We did not detect any particular clustering around dermal adipocytes. The presence of parasites in the skin was not associated with major inflammation (*Supplementary file 1*). To confirm that skin invasion by this parasite was not strain or sub-species specific, the more virulent TREU927 strain of *T. brucei* and the human-infective *T.b. gambiense* strain, PA, were used to infect mice. Extravascular skin invasion of the dermis, subcutaneous adipose tissue and fascial planes (*Figure 2—figure*

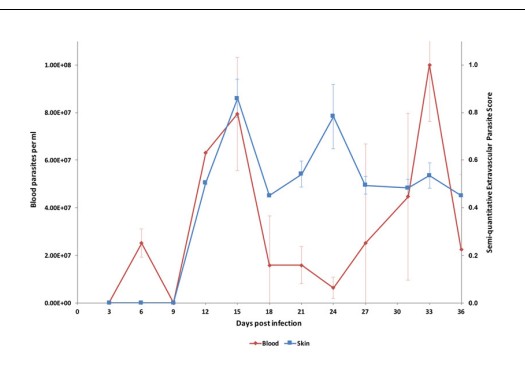

**Figure 1.** Parasite densities in the blood and in the extravascular tissue of the skin over a time-course. The blood parasitaemia of *T.b. brucei* strain STIB247 (red) and the semi-quantitative score of extravascular parasites in the skin (blue) are shown over a 36-day time-course following infection in Balb/C mice. Blood parasitaemia was measured using phase microscopy using methodology outlined in (*Lumsden, 1963*). Skin parasite burden is an average of five high-power fields scored by histological analysis (0 = no parasites detectable; 1 = low numbers of parasites; 2 = moderate numbers of parasites; 3 = large numbers of parasites). Standard error shown (n = 3).

The following source data is available for figure 1:

**Source data 1.** Semi-quantitative evaluation of the parasite burden in skin sections (STIB247) Every three days for 36 days of a STIB247 T.b.

**Source data 2.** Daily parasitaemia during STIB247 infection in Balb/C mice The daily parasitaemia during a 36-day STIB247 T.b.

*supplement 1*) was evident with associated mild to moderate inflammation (*Supplementary file 2*), in some instances the degree of skin invasion in the *T.b. gambiense* infected mice was far greater than *T.b. brucei*, perhaps suggesting a greater propensity for sequestration in this sub-species.

To confirm that the extravascular distribution of parasites was not an artefact of the route of inoculation, infections by natural vector transmission were carried out using a bioluminescent *T.b. brucei* strain, AnTat1.1E AMLuc/tdTomato. Mice were infected by a single infective bite of an individual *G. m. morsitans*. After 4 to 11 days and up to the end of the experiment, parasites were observed in the skin with a dynamic distribution (*Figure 3A*) and a variable density (*Figure 3B*). Parasites were first detected in the blood between 5 and 19 days after natural transmission and parasitaemia remained lower than 10 (*Kanmogne et al., 1996*) parasites/ml. Observed bioluminescence directly reflects the total number of living parasites in the entire organism, including blood and viscera, but the intensity of the signal decreases with tissue depth. Therefore, at the end of each experiment, mice were sacrificed and their organs were checked for bioluminescence. The presence of extravascular parasites in cutaneous and subcutaneous tissues was first demonstrated by bioluminescence imaging in entire dissected skins (*Figure 3C*) and was not necessarily localised to the bite site. In addition to the skin, only the spleen, some lymph nodes and adipose tissue were observed to be positive for bioluminescence in several individuals (*Figure 3—figure supplement 1*). This suggests that the observed bioluminescence is likely to originate predominantly, but not solely, from parasites in the skin. In addition, only mild inflammation was observed after 29 days (*Figure 3—figure supplement 2*).

To confirm that trypanosomes in the skin are a viable population, fluorescent parasites were monitored by two intravital imaging methods following IP injection and natural transmission (*Figure 4*). First, IP-injected fluorescent *T. brucei* STIB247 were imaged in vivo using 2-photon microscopy (*Figure 4A*). Extravascular trypanosomes observed in the dermal layer of dorsal skin were highly motile, consistent with viability (*Video 1*). Second, naturally-transmitted fluorescent *T. brucei* AnTat1.1E AMLuc/tdTomato were imaged in vivo using spinning-disk confocal microscopy in the C57BL/6J-Flk1-EGFP mouse line that has green fluorescent endothelial cells in the lymphatic and

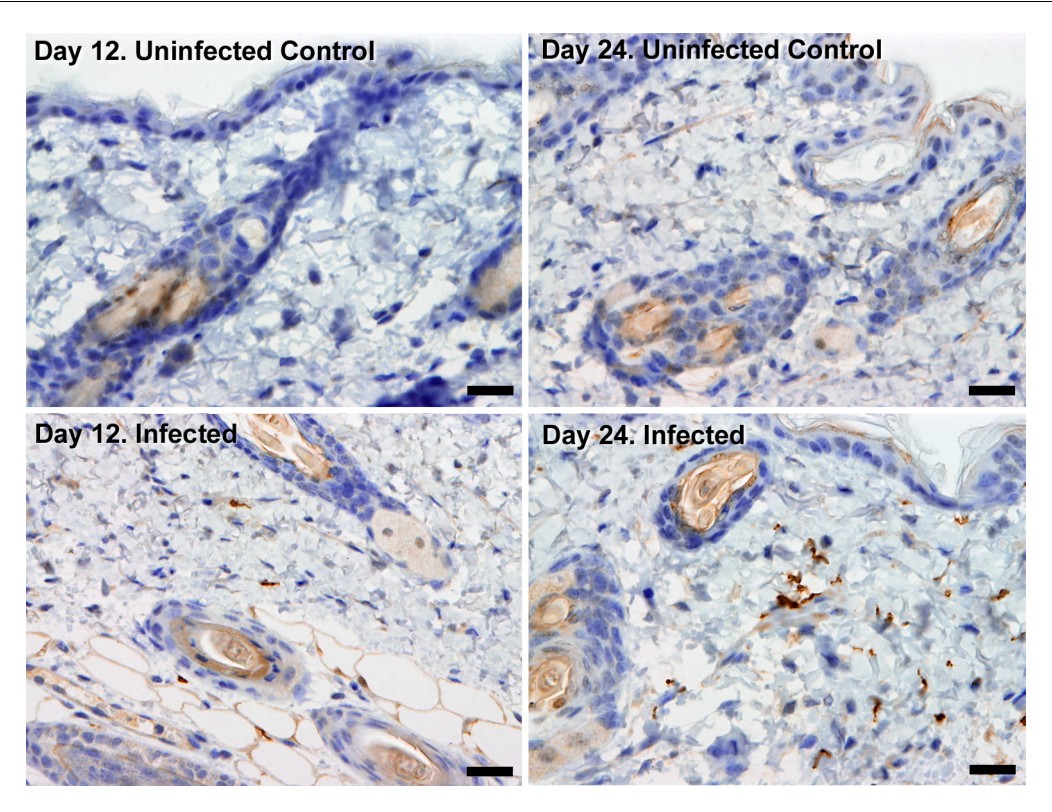

**Figure 2.** Extravascular localisation of trypanosomes during an infection. Histological sections of dorsal skin from uninfected and infected Balb/C mice stained with trypanosome-specific anti-ISG65 antibody (brown), counterstained with Gill's Haematoxylin stain (blue) at 12 days and 24 days post-inoculation with T.b. *brucei* strain STIB247. Parasites are visible in extravascular locations of the skin including the deep dermis and subcutaneous adipose tissue from day 12. The scale bar represents 20 μm.

The following figure supplement is available for figure 2:

**Figure supplement 1.** Skin invasion by T.b. brucei strain TREU927 and T.b. gambiense strain PA.

blood vessels (*Ema et al., 2006*) (*Figure 4B*). Extravascular trypanosomes were observed in the dermal layer of the ear and were highly motile (*Videos 2*, *3* and *4*).

Differentiation of dividing trypanosomes to non-dividing stumpy forms is essential for transmission to the tsetse. Using the relative transcript abundance of the stumpy marker Protein Associated with Differentiation 1 (PAD1) (*Dean et al., 2009*) to an endogenous control, Zinc Finger Protein 3 (ZFP3) (*Walrad et al., 2009*), we estimated that approximately 20% of skin-dwelling parasites were stumpy (*Supplementary file 3*). To directly determine the proportion of stumpy forms in skin sections, histological staining for PAD1 was also performed (*Table 1—source data 4*). PAD1-positive cells were observed in variable proportions (from 8 to 80%) in all bioluminescent skin samples examined after IP injection (*Supplementary file 4*). Following natural transmission, up to 38% of parasites detected using VSG surface markers (*Figure 3—figure supplement 3A–B*) also expressed PAD1 (*Figure 3—figure supplement 3C–D*) (n = 441 cells from 8 skin sections). In all skin sections, stumpy parasites were homogenously distributed in the dermis and subcutaneous adipose tissues.

We next assessed the ability of skin-dwelling parasites to infect tsetse flies. Teneral flies (immature flies that have not yet taken a blood meal) were fed on different regions of skin from mice infected with AnTat1.1E AMLuc/TY1/tdTomato with differing levels of bioluminescence across the skin (*Table 1—source data 1* and *Table 1—source data 2*). This was repeated in 20 mice with differing levels of parasitaemia. Flies were dissected and checked for the presence of fluorescent trypanosomes after two days (*Table 1*). In mice with undetectable or low parasitaemia (<5 × 10^4 parasites/

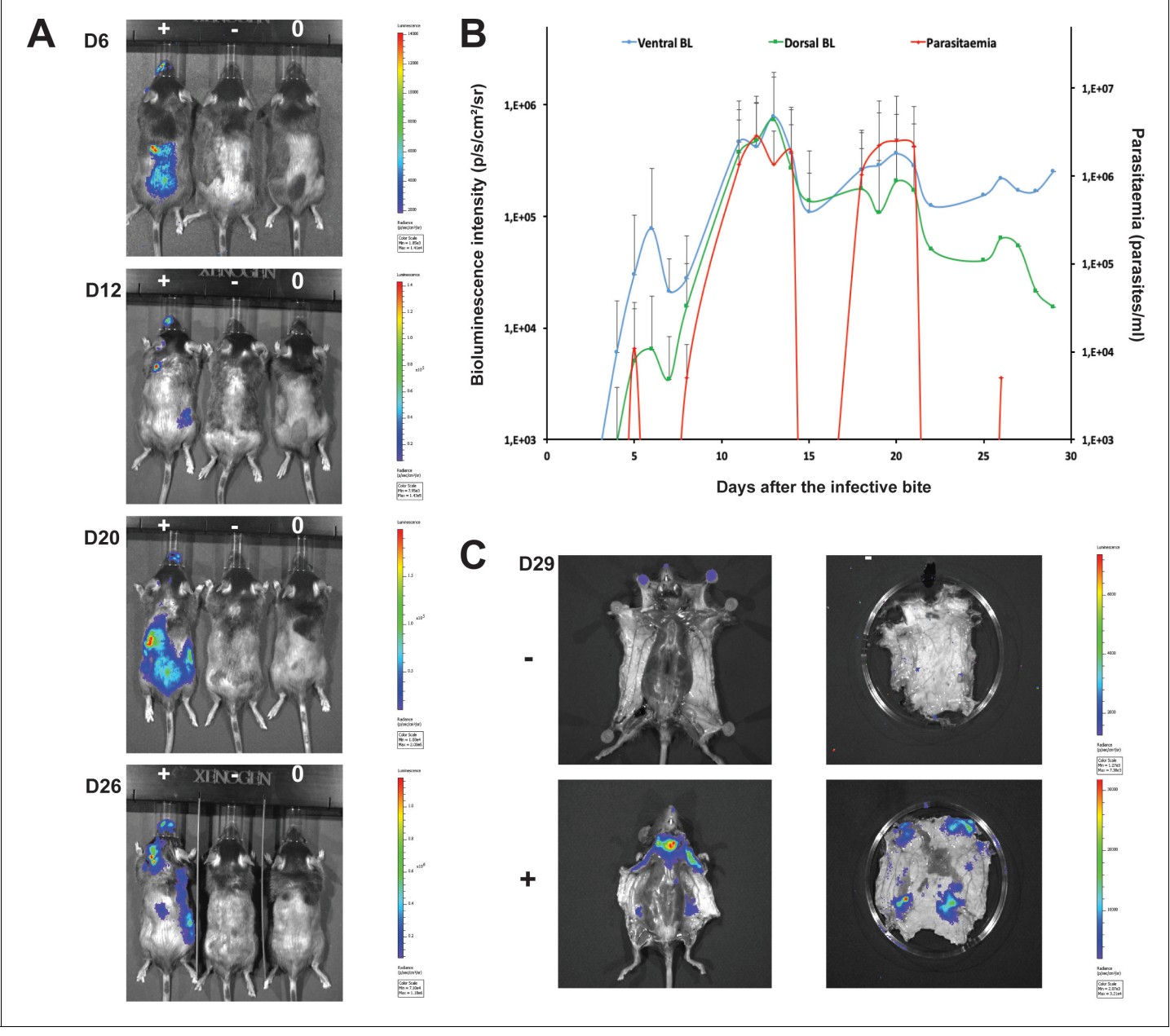

**Figure 3.** Dynamics of parasite distribution in the extravascular tissue of the skin and in the blood during a representative course of infection following natural transmission. A total of seven mice were infected by the single infective bite of an individual G.m. *morsitans* on the belly with the *T.b. brucei* AnTat1.1E AMLuc/tdTomato strain. Panels **A** and **C** depict representative patterns. (**A**) Examples of bioluminescence profiles of 3 mice (+ bitten by an infected fly, - bitten by an uninfected fly and 0 not bitten) 6, 12, 20 and 26 days after the bite are shown. (**B**) Ventral (blue) and dorsal (green) bioluminescence (BL) intensities (in p/s/cm$^2$/sr on the left Y-axis) and parasitaemia (in parasites/ml in red on the right Y-axis) were measured daily for 29 days and plotted as mean ± SD (n = 7 mice). (**C**) The entire skins of mice (+) and (-) were dissected for bioluminescence imaging 29 days after the bite. For the mouse (+), *Figure 3—figure supplement 1* shows the bioluminescence profile of dissected organs, *Figure 3—figure supplement 2* presents the skin inflammation, and *Figure 3—figure supplement 3* shows labelled parasites in skin sections.

The following figure supplements are available for figure 3:

**Figure supplement 1.** Bioluminescence mostly originates from parasites in the skin.

**Figure supplement 2.** Mild inflammation of skin tissues one month after an infection by natural transmission.

**Figure supplement 3.** Extravascular parasites in the skin express both VSGs and PAD1 surface markers.

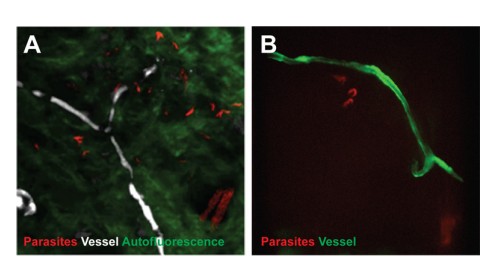

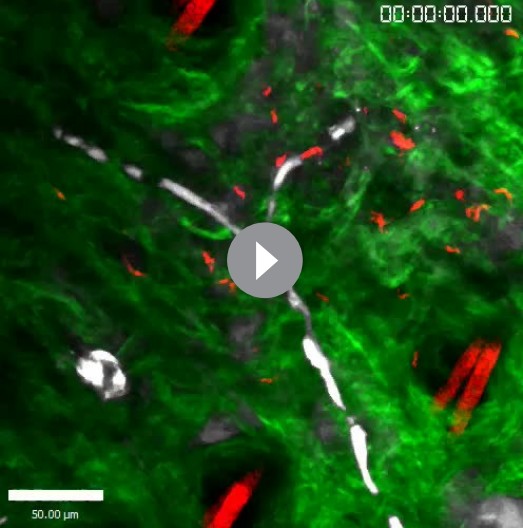

**Figure 4.** Extravascular localisation of trypanosomes during an infection visualised using multi-photon microscopy (**A**) and spinning-disk confocal microscopy (**B**). (**A**) Still-image extracted from video (**Video 1**) of multi-photon live imaging of dorsal skin during a trypanosome infection. Intravenous non-targeted quantum dots (white) highlight blood vessels. *T.b. brucei* STIB 247 parasites transfected with mCherry to aid visualisation (red) are clearly visible and motile outside the vasculature and within the extravascular skin matrix (green). (**B**) Still-image extracted from (**Video 3**) of spinning-disk confocal live imaging of the ear of an *Kdr (Flk1)* C57BL/6J Rj mouse during a trypanosome infection. *T.b. brucei* AnTat1.1E AMLuc/tdTomato parasites expressing tdTomato (red) are moving in the extravascular region surrounding a vessel of the dermis (green).

**Video 1.** Extravascular trypanosomes visualised in the skin using 2-photon microscopy. Intravital multi-photon imaging of the flank skin during trypanosome infection 10 days after IP inoculation. An intravenous injection of non-targeted quantum dots prior to imaging allowed visualisation of blood vessels. mCherry STIB247 *T.b. brucei* parasites (red) are observed moving in the extravascular region surrounding blood vessels of the dermis (white). Collagen auto-fluorescence is visible as green.

ml), no parasites were found in flies fed on non-bioluminescent regions. Conversely, a median of 36% (± 23%, n = 70) of flies that fed on bioluminescent regions of low parasitaemic mice became infected (*Table 1* and *Table 1—source data 3*). This demonstrates that skin parasites, from mice without visible parasitaemia, can contribute to tsetse infection, possibly reflecting human asymptomatic infections. When parasites were detected in the blood and skin, tsetse infection rates increased up to 100% (median of 61% ± 22%, n = 120) (*Table 1* and *Table 1—source data 3*). Parasites taken-up from the skin were able to further develop through the life-cycle to early procyclic forms in the fly as evidenced by GPEET-procyclin marker on the parasites surface (n = 721 cells from 16 flies) (*Table 1—source data 4* and *Supplementary file 4*).

To demonstrate that skin invasion by trypanosomes can occur in humans as well as the murine model, slides of historical skin biopsies, taken as part of a diagnostic screening programme for *Onchocerca* microfilaria in the trypanosomiasis-endemic Democratic Republic of the Congo, were examined for trypanosomes. At the time of sampling, the incidence of trypanosomiasis was 1.5–2% and we hypothesised that some individuals may have harboured undiagnosed infections. Re-examination of 1121 skin biopsies by microscopy revealed six individuals with trypanosomes in their skin (*Figure 5*). These individuals had not previously been diagnosed with human trypanosomiasis by clinical signs or the presence of blood parasites.

## Discussion

We have shown that there exists a significant yet overlooked population of live, motile, extravascular *T. brucei* in the dermis and subcutis of animal models infected by artificial routes or by vector transmission. It is likely that once injected the parasites disseminate via the lymph and blood to the skin where they are ingested during tsetse fly pool-feeding and readily initiate cyclical development in the vector. Given the relative volume of the skin organ compared to the vasculature, it is possible, depending on the density of skin invasion, that more parasites exist in the skin than in the blood.

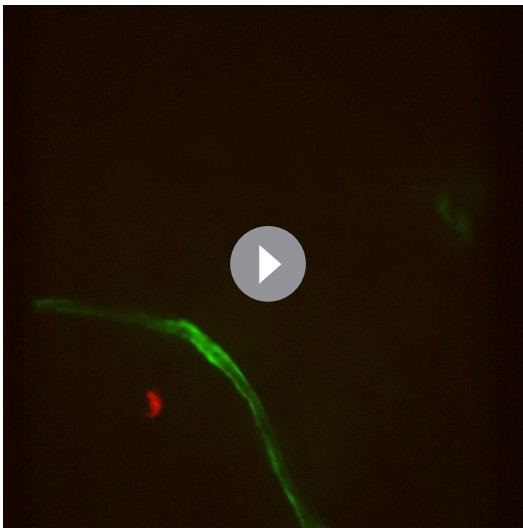

**Video 2.** Extravascular trypanosomes visualised in the skin using spinning-disk confocal microscopy. Spinning-disk confocal live imaging of the ear of an *Kdr (Flk1)* C57BL/6J Rj mouse during a trypanosome infection after natural transmission. *T.b. brucei* AnTat1.1E AMLuc/tdTomato parasites expressing tdTomato (red) are observed moving in the extravascular region surrounding blood and / or lymphatic vessels of the dermis (green).

The skin, therefore, represents an unappreciated reservoir of infection. The extravasation of trypanosomes was described previously in major organs such as liver, spleen (*Blum et al., 2008*; *Kennedy, 2004*) and visceral adipose tissue (*Trindade et al., 2016*), but the importance of these parasites in transmission was not investigated. Here we show that these skin-dwelling trypanosomes contribute to transmission and could explain the maintenance of disease foci, despite active screening and treatment. Skin invasion for enhanced transmission is likely a powerful evolutionary force driving extravasation, suggesting that the generalised tissue penetration underlying pathogenesis (i.e. splenomegaly, hepatomegaly, CNS invasion) is a secondary epiphenomenon. A skin reservoir also presents a novel target for diagnostics (e.g. skin biopsies), allowing the prevalence of infection to be accurately determined and the identification of any previously undetected animal reservoirs of human disease.

The skin as an anatomical reservoir of parasites is a recurring theme in arthropod-borne human diseases such as *Leishmania* (*Sacks, 2008*; *Schlein, 1993*) and *Onchocerca* (*Symptomatology, 1974*; *Dalmat, 1955*). Here we present evidence of trypanosomes in the skin of hitherto undiagnosed individuals. This anatomical reservoir may serve to explain how HAT foci re-emerge and persist despite low numbers of reported cases even in the absence of an animal reservoir (*Kagbadouno et al., 2012*; *Cordon-Obras et al., 2009*; *Balyeidhusa et al., 2012*).

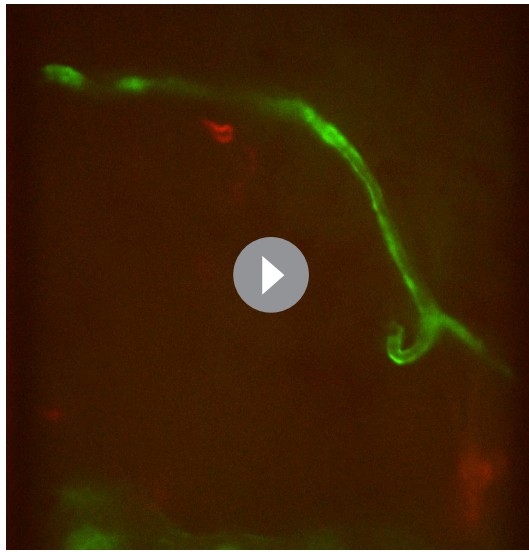

**Video 3.** Extravascular trypanosomes visualised in the skin using spinning-disk confocal microscopy.

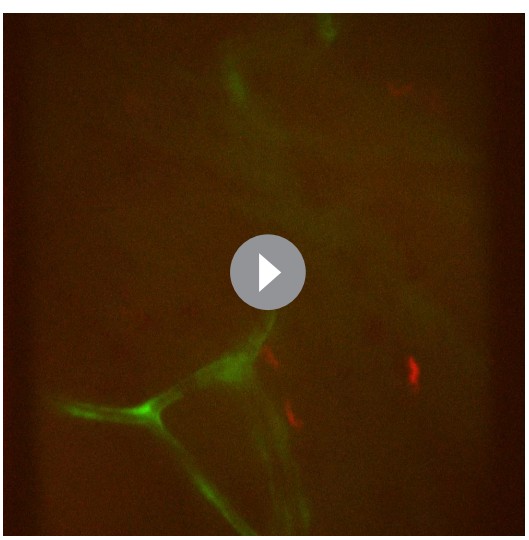

**Video 4.** Extravascular trypanosomes visualised in the skin using spinning-disk confocal microscopy.

**Table 1.** Skin parasites are ingested during tsetse pool-feeding. Mice were IP infected with *T.b. brucei* AnTat1.1E AMLuc/TY1/tdTomato and the parasitaemia and bioluminescence were monitored daily until the day of xenodiagnosis. The number of parasites in the blood was determined using a haemocytometer or a flux cytometer. The number of parasites in the skin was estimated from the measured bioluminescence intensity by using a standard curve (**Table 1—source data 1** and **Table 1—source data 2**). Batches of teneral flies were fed on different skin regions of mice infected with differing levels of bioluminescence across the skin and with differing levels of parasitaemia (**Table 1—source data 2** and **Table 1—source data 3**). Fly batches A4A, A2A, B1A, 3B and 3A were used to assess tsetse transmission in hosts with low numbers of blood parasites but high numbers skin parasites, while fly batches 1B, 1A, 4B, 4A, 2B, 2A, B2B and B4A were used to investigate the impact of high numbers of parasites in both the skin and blood. Flies were dissected and their midguts checked for the presence of fluorescent trypanosomes after two days to determine the proportion of infected flies (**Table 1—source data 4A–B**). For some of these experiments, results of an in-depth quantification of parasite stages by IFA is provided in **Supplementary file 4**. Stumpy forms were observed only in the blood of mice with parasitaemia values highlighted in purple. Bioluminescence was detected in the skin of mice with values highlighted in grey.

| Fly batches | Parasites in blood (per ml) | Parasites in skin (per cm$^2$) | Dissected flies | Fly infection rates (%) |
|---|---|---|---|---|
| C1 | 0 | 0 | 32 | 0% |
| C | 0 | 0 | 8 | 0% |
| A4B | $< 10^4$ | $< 10^3$ | 16 | 0% |
| B1B | $2.2 \times 10^4$ | $< 10^3$ | 13 | 0% |
| A4A | $< 10^4$ | $6.6 \times 10^5$ | 17 | 35% |
| A2A | $1.1 \times 10^4$ | $3.8 \times 10^6$ | 7 | 86% |
| B1A | $2.2 \times 10^4$ | $4.6 \times 10^7$ | 16 | 31% |
| 3B | $4.4 \times 10^4$ | $2.6 \times 10^4$ | 14 | 36% |
| 3A | $4.4 \times 10^4$ | $2.6 \times 10^4$ | 16 | 38% |
| 1B | $1.8 \times 10^5$ | $8.0 \times 10^3$ | 12 | 67% |
| 1A | $1.8 \times 10^5$ | $8.0 \times 10^3$ | 14 | 79% |
| 4B | $2.2 \times 10^5$ | $1.2 \times 10^4$ | 17 | 53% |
| 4A | $2.2 \times 10^5$ | $1.2 \times 10^4$ | 18 | 56% |
| 2B | $1.6 \times 10^6$ | $8.0 \times 10^3$ | 14 | 36% |
| 2A | $1.6 \times 10^6$ | $3.2 \times 10^4$ | 18 | 39% |
| B4B | $4.3 \times 10^6$ | $6.7 \times 10^7$ | 10 | 80% |
| B4A | $4.3 \times 10^6$ | $6.7 \times 10^7$ | 17 | 100% |

Source data 1. Characterisation of the AnTat 1.1E AMLuc/TY1/tdTomato sub-clone. (A) The in vitro growth of the selected AnTat1.1E AMLuc/TY1/tdTomato sub-clone (red) was similar to that of the parental wild-type strain (blue). Bloodstream forms were cultured in HMI11, counted daily in a Muse cytometer (Merck-Millipore) and diluted after 4 days. (B) A parasite density / bioluminescence intensity analysis was performed by measuring the bioluminescence in successive 2-fold dilutions in 96-micro-well plates with an IVIS Spectrum imager (Perkin Elmer). When plotted as mean ± SD (n = 3), parasite densities and bioluminescence intensities were correlated when the bioluminescence levels were higher than 10 (Berthier et al., 2016) p/s/cm$^2$/sr, corresponding to about 10 (**Koffi et al., 2006**) parasites, allowing estimation of the parasite density from in vivo imaging over this threshold. This standard curve was used to estimate the number of parasites in the skin from measured values of bioluminescence. (C) This correlation was verified by quantification in a microplate reader Infinite 200 (Tecan) at the very beginning of the first in vivo experiment as well as the end of the last one (mean ± SD, n = 3).

Source data 2. Parasite densities in extravascular tissue of the skin and in the blood of mice used for differential xenodiagnosis. Mice were injected IP with AnTat1.1E AMLuc/TY1/tdTomato and monitored daily for bioluminescence and parasitaemia. (A) Bioluminescence profile of four mice (- uninfected control and (1–3) three infected mice) four days after infection. (B) The entire skins of the uninfected control mouse (-) and mouse 3 were dissected for bioluminescence imaging four days after infection. (C) Parasite densities in the blood and in the skin (calculated from the mean dorsal bioluminescence intensity measurement and from the standard curve in **Source data 1**, in parasites/cm (**Fakhar, 2013**) in blue) were calculated daily over one week and plotted as mean ± SD (n = 13 mice).

Source data 3. Skin parasites are sufficient to initiate a tsetse infection. Schematics summarising the principal results from the xenodiagnosis experiment. In a mouse with no detected transmissible parasites in the blood (absence of stumpy forms by IFA and absence of infection of flies fed on a non-bioluminescent region of the skin), flies can ingest transmissible parasites from the bioluminescent region of the skin (left panel). When a mouse

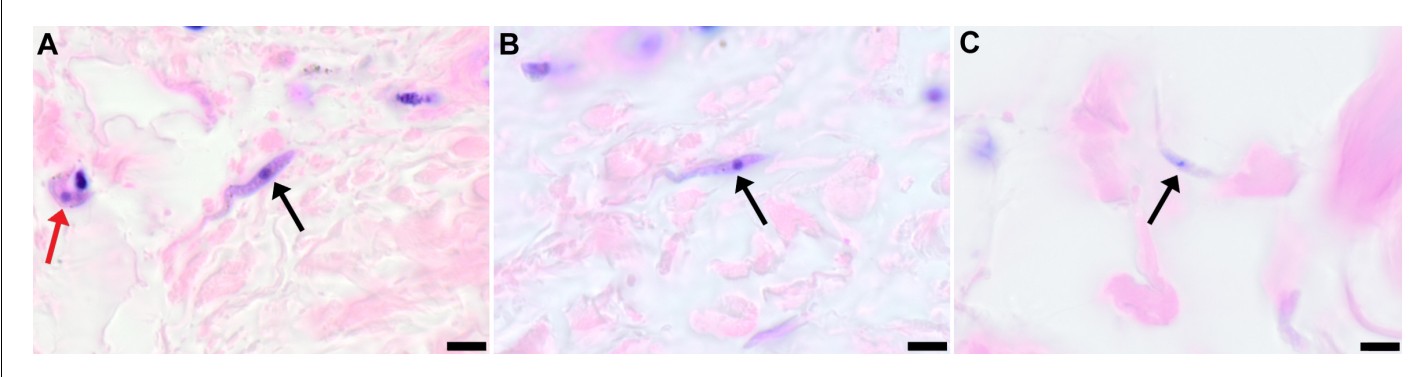

**Figure 5.** Extravascular localisation of trypanosomes in previously unidentified human cases of trypanosomiasis Histological sections of skin collected from previously unidentified cases of human trypanosomiasis from the Democratic Republic of Congo, showing the presence of extravascular parasites in biopsies from three individuals (A, B and C). Skin biopsies were collected as part of a national onchocerciasis screening programme that took place in the same geographic region as an active trypanosomiasis focus. Slides were stained with Giemsa and examined under oil immersion at 100x magnification. In addition to visible slender forms (black arrows) in the extravascular tissue of the skin, a clearly identifiable stumpy transmission form with typical morphology and an unattached undulating membrane is also present in the skin of one individual (red arrow in A). The scale bar represents 5 µm.

HAT was once widespread across much of sub-Saharan Africa but concerted control efforts brought it close to elimination during the 1960s (*World Health Organization 2000*, *2013*). However, disease foci persisted, with a resurgence in the number of reported cases to over 300,000 in the 1990s (*World Health Organization 2000*; *Steverding, 2008*). Currently, HAT is approaching elimination in many areas (*World Health Organization 2013*; *Simarro et al., 2012*). Understanding how HAT evaded elimination in the latter half of the 20th century and how it continues to persist is vital to efforts to eliminate the disease. For example, our results indicate that it may be necessary to develop novel therapeutics capable of accessing extravascular compartments. The current policy in most endemic countries is not to treat serologically positive individuals unless they demonstrate active infection, due to the long duration and high toxicity of treatment and the low predictive value of the serological tests. We suggest that this policy should be reconsidered in light of our compelling evidence that they represent a carrier population which may maintain HAT foci and explain previously thwarted efforts to eliminate this major pathogen.

## Materials and methods

### Trypanosome strains and cultures

All strains used in this study are pleomorphic. STIB247 is a low virulence *T.b. brucei* strain that induces a chronic infection and was isolated from a hartebeest in the Serengeti in 1971 (*Geigy et al., 1973*). TREU927 is the genome reference strain for *T. brucei* and is more virulent than STIB247. This strain was isolated from a tsetse fly in Kenya in 1969/1970 (*Goedbloed et al., 1973*). PA is a human-infective group 1 *T.b. gambiense* strain isolated from a patient in the Democratic Republic of the Congo in 1975 (*Tait et al., 1984*). mCherry STIB247 was created by transfection of STIB247 with pHD1034-mCherry (*Myburgh et al., 2013*). This strain constitutively expresses fluorescent mCherry from the ribosomal RNA promoter and its expression is stable over repeated passage.

The AnTat 1.1E clone of *T.b. brucei* was derived from a strain originally isolated from a bushbuck in Uganda in 1966 (*Le Ray et al., 1977*). Bloodstream forms were cultivated in HMI11 medium supplemented with 10% foetal calf serum at 37°C in 5% $CO_2$. This strain was genetically engineered to produce two strains continuously expressing the red-shifted luciferase (PpyRE9H) and the tdTomato red fluorescent protein either individually or in combination.

For the first AnTat 1.1E strain, the pTbAMLuc plasmid (M. Taylor, London School of Hygiene and Tropical Medicine, UK) was used for continuous cytosolic expression of the red-shifted luciferase PpyRE9H. The tdTomato coding sequence was cloned with *Hind*III and *Bam*HI into the pTSARib

vector (*Xong et al., 1998*), generating the final pTSARib-tdTomato construct. The two plasmids were linearised with *Kpn*I and *Sph*I restriction enzymes, respectively, and used to transform procyclic parasites with an Amaxa Nucleofector (Lonza) (*Burkard et al., 2007*). After 24 hr, transfected cells were selected by the addition of blasticidin or puromycin (10 µg/ml). After one week, the population was examined (i) for red fluorescence by fluorescence microscopy and (ii) for both red fluorescence and bioluminescence by using a fluorimeter Infinite 200 (Tecan, Switzerland). Cells were sub-cloned by limiting dilution, and clone selection was performed after 15 days by measuring both bioluminescence in a microplate reader Infinite 200 (Tecan) and fluorescence with a Muse cell Analyzer (Merck-Millipore). This strain, named AnTat1.1E AMLuc/tdTomato, was used for natural transmission experiments.

The second AnTat 1.1E strain expressing the 3.1 Kb chimeric multiplex reporter protein PpyRE9H::TY1::tdTomato was named AnTat 1.1E AMLuc/TY1/tdTomato. This cytoplasmic reporter is composed of three distinct markers: the red-shifted luciferase (PpyRE9H) is fused to the tdTomato red fluorescent protein by a TY1 tag. Briefly, the 1.68 Kb optimised version of the North American firefly *Photinus pyralis* luciferase (*Branchini et al., 2005*) was fused with a 10-bp sequence known as TY1-tag (*Bastin et al., 1996*) and cloned into the pTSARib plasmid (*Xong et al., 1998*) by using *Xho*I and *Hin*dIII restriction enzymes to obtain the pTSARib-PpyRE9H-TY1 plasmid. Finally, the 1.4 Kb coding region of the tdTomato fluorescent protein was inserted downstream using *Hin*dIII and *Bam*HI. The resulting 8.9 Kb vector, containing a blasticidin S resistance cassette, was linearised with *Sph*I to integrate the rDNA promoter locus. Bloodstream parasites were transformed with an Amaxa Nucleofector (Lonza) (*Burkard et al., 2007*), sub-cloned by limiting dilution, and clone selection was performed by measuring both bioluminescence in a microplate reader Infinite 200 (Tecan) and fluorescence with a Muse cell Analyzer (Merck-Millipore). The selected AnTat 1.1E AMLuc/TY1/tdTomato sub-clone was comparable to the parental wild-type strain in terms of growth rate (*Table 1—source data 1A*), pleomorphism (*Table 1—source data 4C*), tsetse infectivity and virulence in mice (*Table 1—source data 2*). In order to verify the reliability of the bioluminescent marker as well as to define the bioluminescence detection threshold of the AnTat 1.1E AMLuc/TY1/tdTomato selected sub-clone, a parasite density / bioluminescence intensity analysis was performed in 96-micro-well plates with an IVIS Spectrum imager (Perkin Elmer). Parasite density and bioluminescence intensity were correlated when bioluminescence levels were higher than $10^4$ p/s/cm$^2$/sr, corresponding to about $10^3$ parasites, allowing estimation of the parasite density from in vivo imaging over this threshold (*Table 1—source data 1B*). This correlation was verified by quantification in a microplate reader Infinite 200 (Tecan) at the very beginning of the first in vivo experiment as well as the end of the last one, demonstrating the stability of the triple reporter expression in the AnTat 1.1E AMLuc/TY1/tdTomato selected sub-clone over time, especially after at least one full in vivo parasite cycle in the tsetse fly and the mammalian host (*Table 1—source data 1C*). This strain was used for xenodiagnosis experiments and quantification of parasite densities.

## Mouse strains

BALB/c and C57BL/6J mice were used as models for chronic disease (*Magez and Caljon, 2011*). In addition, to allow for further 3D intravital imaging of the lymphatic and blood systems, C57BL/6J-Flk1-EGFP mice expressing a GFP tagged *Kdr (Flk1)* gene encoding the vascular endothelial growth factor receptor 2 (VEGFR-2) were used (*Ema et al., 2006*).

## Ethical statements

This study was conducted under Home Office and SAPO regulations in the UK and in strict accordance with the recommendations from the Guide for the Care and Use of Laboratory Animals of the European Union (European Directive 2010/63/UE) and the French Government. The protocol was approved by the 'Comité d'éthique en expérimentation animale de l'Institut Pasteur' CETEA 89 (Permit number: 2012–0043 and 2016–0017) and undertaken in compliance with Institut Pasteur Biosafety Committee (protocol CHSCT 12.131).

## Skin invasion time-course

A total of 36 eight-week old BALB/c mice (Harlan, UK) were inoculated by intra-peritoneal (IP) injection with $10^4$ parasites of strain STIB 247. Parasitaemia was assayed on each subsequent day using

phase microscopy (*Lumsden, 1963*). Twenty-four uninfected animals served as controls. Every three days for 36 days, three infected animals and two uninfected animals were culled and 2 cm$^2$ skin samples removed from the dorsum. Skin samples were fixed in 10% neutral buffered formalin prior to histological analysis.

## Natural infections using tsetse flies

Tsetse flies (*Glossina morsitans morsitans)* were maintained, infected and dissected at the Institut Pasteur as described previously (*Rotureau et al., 2012*). Flies were infected with AnTat1.1E AMLuc/tdTomato parasites. Positive flies were selected first by screening the abdominal fluorescence (midgut infection) 15 days after the infective meal, and then by a salivation test (mature salivary gland infection) after one month. Single flies with salivary gland infections were used to infect the abdomen of mice anaesthetised by IP injection of ketamine (Imalgene 1000 at 125 mg/kg) and xylazine (Rompun 2% at 12.5 mg/kg) and feeding was confirmed by visual observation of the fly abdomen full of blood. Control mice were either not bitten or bitten by uninfected flies. The presence and density of parasites in the blood was determined daily by automated fluorescent cell counting with a Muse cytometer (Merck-Milllipore, detection limit 5.10$^2$ parasites/ml) or by direct examination under a phase microscope with standardised one-use haemocytometers (Hycor Kova, detection limit 10$^4$ parasites/ml), according to the manufacturer's recommendations.

## Infection for xenodiagnosis

A total of 20 seven-week-old male C57BL/6J Rj mice (Janvier, France) were IP injected with 10 (*Jamonneau et al., 2012*) AnTat 1.1E AMLuc/TY1/tdTomato bloodstream forms. Parasitaemia was assayed daily by automated fluorescent cell counting with a Muse cytometer (Merck-Millipore, detection limit 5.10$^2$ parasites/ml) according to the manufacturer's recommendations.

## PAD1/ZFP3 relative expression

Three BALB/c were infected with *T.b. brucei* strain TREU927 and culled at day 11. The mice were perfused and a 2 cm$^2$ region of skin removed from the flank. Skin sections were lysed using a Qiagen Tissuelyzer LT and RNA extracted using a Qiagen RNAeasy kit following the manufacturer's instructions. 100 ng of RNA from each sample was reverse-transcribed using an Invitrogen Superscript III RT kit. qPCR was performed on each sample using 5 µl of cDNA using a protocol and primers validated previously (*MacGregor et al., 2011*) on an Agilent Technologies Stratagene Mx3005P qPCR machine. The ratio of *PAD1* to *ZFP3*, and hence the proportion of cells transcribing the *PAD1* gene, was estimated using the Agilent Technologies MXPro software.

## Xenodiagnosis

Mice were first anaesthetised by IP injection of ketamine (Imalgene 1000 at 125 mg/kg) and xylazine (Rompun 2% at 12.5 mg/kg). Batches of 10 teneral male tsetse flies (from 8 to 24 hr post-eclosion) were then placed in 50 ml Falcon tubes closed with a piece of net through which they were allowed to feed directly on mouse skin regions of interest for 10 min. The selection of the skin regions for fly feeding was based on mice bioluminescence profiles and parasitaemia. Unfed flies were discarded and fed flies were maintained as previously described. Anaesthetised mice were finally sacrificed by cervical dislocation and their skin was dissected for controlling bioluminescence with an IVIS Spectrum imager (Perkin Elmer). All the flies were dissected and checked for the presence of trypanosomes either 2 or 14 days after their meal on mouse skin by two entomologists blinded to group assignment and experimental procedures. Dissections were performed as previously described (*Rotureau et al., 2012*), entire midguts were scrutinised by fluorescence microscopy to detect and count living red fluorescent parasites, and positive midguts were further treated for IFA. A total of 420 flies were used in 3 independent xenodiagnosis experiments.

## In vitro bioluminescence imaging

To perform the parasite density / bioluminescence intensity assay with AnTat 1.1E AMLuc/TY1/tdTomato bloodstream forms, parasites were counted, centrifuged and resuspended in fresh HMI11 medium at 10.10$^6$ cells/ml. Then, 100 µl (or 10$^6$ parasites) of this suspension were transferred into black clear-bottom 96-well plates and serial 2-fold dilutions were performed in triplicate adjusting

the final volume to 200 μl of HMI11 with 300 μg/ml of beetle luciferin (Promega, France). Luciferase activity was quantified after 10 min of incubation with a microplate reader Infinite 200 (Tecan), following the instructions of the Promega Luciferase Assay System. After background removal, results were analysed as mean ± SD of three independent experiments.

## In vivo bioluminescence imaging

Infection with bioluminescent parasites was monitored daily by detecting the bioluminescence in whole animals with the IVIS Spectrum imager (Perkin Elmer). The equipment consists of a cooled charge-coupled camera mounted on a light-tight chamber with a nose cone delivery device to keep the mice anaesthetised during image acquisition with 1.5% isofluorane. D-luciferin potassium salt (Promega) stock solution was prepared in phosphate buffered saline (PBS) at 33.33 mg/ml, filter-sterilised and stored in a −20°C freezer. To produce bioluminescence, mice were inoculated IP with 150 μl of D-luciferin stock solution (250 mg/kg). After 10 min of incubation to allow substrate dissemination, all mice were anaesthetised in an oxygen-rich induction chamber with 2% isofluorane, and images were acquired by using automatic exposure (30 s to 5 min) depending on signal intensity. Images were analysed with Living Image software version 4.3.1 (Perkin Elmer). Data were expressed in average radiance (p/s/cm$^2$/sr) corresponding to the total flux of bioluminescent signal according to the selected area (total body of the mouse here). The background noise was removed by subtracting the bioluminescent signal of the control mouse from the infected ones for each acquisition.

## 2-photon microscopy

Intravital multi-photon microscopy studies were carried out using a Zeiss LSM7 MP system equipped with a tuneable titanium:sapphire solid-state two-photon excitation source (4W, Chameleon Ultra II, Coherent Laser Group) coupled to an Optical Parametric Oscillator (Chameleon Compact OPO; Coherent). Movies were acquired for 10 to 15 min with an X:Y pixel resolution of 512 × 512 in 2 μm Z increments producing up to 40 μm stacks. 3D tracking was performed using Volocity 6.1.1 (Perkin Elmer, Cambridge, UK). Values representing the mean velocity, displacement and meandering index were calculated for each object. Mice were anaesthetised IP using medetomidine (Domitor 0.5 mg/kg) and ketamine (50 mg/kg) and placed on a heated stage. Following removal of hair with a depilatory cream, dorsal skin was imaged. An intravenous injection of non-targeted quantum dots (Qdot705) (Life Technologies, UK) prior to imaging allowed visualisation of blood vessels.

## Spinning-disk confocal microscopy

AnTat1.1E AMLuc/tdTomato parasites were monitored in the ear of *Kdr (Flk1)* C57BL/6J Rj mice by spinning-disk confocal microscopy as described previously (*Rotureau et al., 2012*). Briefly, mice were first anaesthetised by IP injection of ketamine (Imalgene 1000 at 125 mg/kg) and xylazine (Rompun 2% at 12.5 mg/kg). Mice were wrapped in a heating blanket and placed on an aluminium platform with a central round opening of 21 mm in diameter. A coverslip was taped on the central hole and the mouse was positioned so that the ear was lying on this oiled coverslip. Imaging was performed using an UltraView ERS spinning-disk confocal system (Perkin Elmer) with a x40 oil objective (1.3 numerical aperture). Movies were acquired by an EM-CCD camera (Hamamatsu) controlled by the Volocity software (Perkin Elmer) with an exposure time of 500 ms for a total of 30 to 120 s. Images were analysed using ImageJ 1.48v and its plugin Bio-formats importer (NIH).

## Histological and immunohistochemical evaluation of the skin

Paraformaldehyde-fixed skin samples were trimmed and processed into paraffin blocks. Sections were stained with Haematoxylin and Eosin (HE). Additional serial sections were processed for immunohistochemical staining using a polyclonal rabbit antibody raised against the invariant surface glycoprotein 65 (IGS65) (M. Carrington, Cambridge, UK) using a Dako Autostainer Link 48 (Dako, Denmark) and were subsequently counterstained with Gill's Haematoxylin.

## Histopathological assessment of inflammation in the skin

The extent of cutaneous inflammatory cell infiltration was assessed in haematoxylin and eosin stained sections with a semi-quantitative scoring system applied by two pathologists blinded to group

assignment and experimental procedures. The extent of mixed inflammatory cell infiltration in the dermis and/or subcutis was assessed on a 0 to 3 grading scale (0 = no inflammation or only few scattered leukocytes; 1 = low numbers of inflammatory cells; 2 = moderate numbers of inflammatory cells; 3 = large numbers of inflammatory cells). Ten high-power fields were scored for each skin sample. An inflammation score calculated as the average of the scores in the 10 high-power fields was determined for each sample.

### Semi-quantitative evaluation of the parasite burden in skin sections

Parasite burden was assessed in skin sections stained with anti-IGS65 antibody by two pathologists blinded to group assignment and experimental procedures. Presence of parasites defined as intravascular (parasites within the lumen of dermal or subcutaneous small to medium-sized vessels) and extravascular (parasites located outside blood vessels, scattered in the connective tissue of the dermis or in the subcutis) was evaluated in 5 randomly selected high-power fields at x40 magnification with a 0 to 3 semi-quantitative grading scale (0 = no parasites detectable; 1 = low numbers of parasites (<20); 2 = moderate numbers of parasites (20 < 50); 3 = large numbers of parasites (>50). An average parasite burden score was calculated for each sample.

### Immunofluorescence analysis

Cells were treated for immunofluorescence after paraformaldehyde or methanol fixation as described previously (*Dean et al., 2009*). Parasites were stained with one or two of the following antibodies: (i) the anti-CRD polyclonal rabbit antibody (1:300) to label the cross-reactive determinant of the glycosylphosphatidylinositol anchors of proteins, predominantly the variant surface glycoproteins (*Zamze et al., 1988*), (ii) the anti-PAD1 polyclonal rabbit antibody (1:100) targeting the carboxylate-transporter Proteins Associated with Differentiation 1 (PAD1) (Keith Matthews, Edinburgh, UK) (*Dean et al., 2009*), (iii) the anti-GPEET mouse IgG3 monoclonal antibody (1:500) targeting the *T. brucei* GPEET-rich procyclin (Acris Antibodies GmbH, San Diego, USA), (iv) the L8C4 mouse IgG1 monoclonal antibody labelling an epitope of the PFR2 protein (*Kohl et al., 1999*). Specific antibodies with minimal cross-reactions with mice and coupled to AlexaFluor 488, Cy3 or Cy5 (Jackson ImmunoResearch, USA) were used as secondary antibodies. DNA was stained with 4,6-diamidino-2-phenylindole (DAPI). IFA image acquisition was carried out on a Leica 4000B microscope with a x100 objective lens using a Hamamatsu ORCA-03G camera controlled by Micro-manager and images were normalised and analysed with ImageJ 1.49v (NIH).

### Histopathology of historical human skin samples

Historical human skin samples were collected from 1991 to 1995 as part of The National Onchocerciasis Task Force (NOTF) (*Makenga Bof et al., 2015*). Of this collection, 1121 paraffin embedded skin samples were cut into 2.5 micron sections and stained with Giemsa (Sigma-Aldrich). Slides were screened for the presence of parasites by two pathologists independently and representative images taken at x100 magnification.

## Acknowledgements

We acknowledge P Solano, D Engman, K Matthews, M Taylor, M Carrington, R Amino, E Myburgh and K Gull for providing various material, cell lines, antibodies and plasmids and to A Tait for critically reviewing this manuscript.

## Additional information

### Funding

| Funder | Grant reference number | Author |
| --- | --- | --- |
| Wellcome Trust | Senior Fellowship (Annette Macleod) - 095201/Z/10/Z | Paul Capewell<br>Caroline Clucas<br>William Weir<br>Anneli Cooper<br>Annette MacLeod |

| Wellcome Trust | Wellcome Trust Centre for Molecular Parasitology Core Funding - 085349 | Paul Capewell<br>Caroline Clucas<br>William Weir<br>Anneli Cooper<br>Annette MacLeod |
|---|---|---|
| Agence Nationale de la Recherche | Young Researcher Grant (ANR-14-CE14-0019-01) | Brice Rotureau |
| Agence Nationale de la Recherche | Post-doctoral fellowship (ANR-14-CE14-0019-01) | Estefania Calvo-Alvarez |
| Agence Nationale de la Recherche | Investissement d'Avenir programme, Laboratoire d'Excellence (ANR-10-LABX-62-IBEID) | Paul Capewell |
| Institut Pasteur | | Estefania Calvo-Alvarez<br>Aline Crouzols<br>Grégory Jouvion<br>Brice Rotureau |
| Institut National de la Santé et de la Recherche Médicale | | Estefania Calvo-Alvarez<br>Aline Crouzols<br>Grégory Jouvion<br>Brice Rotureau |

The funders had no role in study design, data collection and interpretation, or the decision to submit the work for publication.

## Author contributions

PC, CC-T, FM, PJ, CC, RAB, ACo, N-rKS, DMN, Acquisition of data, Analysis and interpretation of data, Drafting or revising the article; T-AG, ACr, GJ, MLS, KO, Acquisition of data, Analysis and interpretation of data; EC-A, Acquisition of data, Analysis and interpretation of data; VJ, BB, Acquisition of data; WW, Analysis and interpretation of data, Drafting or revising the article; PG, Conception and design; BR, Conception and design, Acquisition of data, Analysis and interpretation of data, Drafting or revising the article; AM, Conception and design, Analysis and interpretation of data, Drafting or revising the article

## Author ORCIDs

Annette MacLeod, http://orcid.org/0000-0002-0150-5049

## Ethics

Animal experimentation: This study was conducted and licenced under Home Office and SAPO regulations in the UK and in strict accordance with the recommendations from the Guide for the Care and Use of Laboratory Animals of the European Union (European Directive 2010/63/UE) and the French Government. The protocol was approved by the "Comité d'éthique en expérimentation animale de l'Institut Pasteur" CETEA 89 (Permit number: 2012-0043 and 2016-0017) and carried out in compliance with Institut Pasteur Biosafety Committee (protocol CHSCT 12.131).

## Additional files

### Supplementary files

• Supplementary file 1. Histopathological assessment of inflammation in the skin during STIB247 infection. The extent of cutaneous inflammatory cell infiltration during the 36-day STIB247 experiment was assessed on haematoxylin and eosin stained sections with a semi-quantitative scoring system applied by two pathologists blinded to group assignment and experimental procedures.The extent of mixed inflammatory cell infiltration in the dermis and/or subcutis was assessed on a 0 to 3 grading scale (0 = no inflammation or only few scattered leukocytes; 1 = low numbers of inflammatory cells; 2 = moderate numbers of inflammatory cells; 3 = large numbers of inflammatory cells). Ten high-power fields (HPFs) were scored for each skin sample.

• Supplementary file 2. Histopathological assessment of inflammation in the skin during TREU927 infection. The extent of cutaneous inflammatory cell infiltration at day 10 of infection by strain TREU927 experiment was assessed on haematoxylin and eosin stained sections with a semi-quantitative scoring system applied by two pathologists blinded to group assignment and experimental procedures. The extent of mixed inflammatory cell infiltration in the dermis and/or subcutis was assessed on a 0 to 3 grading scale (0 = no inflammation or only few scattered leukocytes; 1 = low numbers of inflammatory cells; 2 = moderate numbers of inflammatory cells; 3 = large numbers of inflammatory cells). Ten high-power fields (HPFs) were scored for each skin sample.

• Supplementary file 3. Expression of PAD1 relative to ZFP3. The relative abundance stumpy cells in the skin of three BALB/c was estimated using qPCR at day 11 post-inoculation with *T.b. brucei* strain TREU927. Mice were culled and perfused to remove blood parasites and a 2 cm (*Fakhar, 2013*) region of skin removed from the flank. The tissue was homogenised and RNA extracted. 100 ng of RNA from each sample was reverse-transcribed and qPCR performed to estimate the cycle thresholds ($C_T$) of the stumpy marker PAD1 and the endogenous control ZFP3. As $C_T$ is inversely proportional to amount of target cDNA in the sample and PAD1 and ZFP3 have similar qPCR efficiencies, a comparison of the delta (Δ) of $C_T$ between PAD1 and ZFP3 transcripts reveals the relative ratio of PAD1 to ZFP3 transcripts and hence the proportion of differentiated parasites transcribing the PAD1 gene.

• Supplementary file 4. Both the respective densities and the proportions of transmissible forms of parasites in the skin and in the blood govern the tsetse infection rates during pool feeding. For some of the xenodiagnosis experiments shown in *Table 1*, identification and quantification of parasite stages was performed by IFA on blood smears and skin sections.Stumpy forms were observed only in the blood of mice with parasitaemia values highlighted in light grey. Bioluminescence was detected in the skin of mice with values highlighted in dark grey. The number of parasites in the skin was calculated according to the values obtained in the standard in vitro assay (*Table 1—source data 1*) and is therefore probably an underestimate. Tsetse flies were dissected 2 days after xenodiagnosis. Populations of intermediate and stumpy form cells were assessed in blood smears and in successive skin sections stained either with the anti-CRD antibody or the anti-PAD1 antibody (see Materials and method section). Populations of early procyclic cells were assessed in dissected fly midguts stained with the anti-GPEET antibody (see Materials and method section). ND: not determined.

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
