## [Decision Letter]

Thank you for submitting your article "The skin is a significant but overlooked anatomical reservoir for vector-borne African trypanosomes" for consideration by *eLife*. Your article has been favorably evaluated by Wendy Garrett (Senior Editor) and three reviewers, one of whom, Photini Sinnis (Reviewer #1), is a member of our Board of Reviewing Editors. The following individual involved in review of your submission has agreed to reveal their identity: Álvaro Acosta Serrano (Reviewer #2).

The reviewers have discussed the reviews with one another and the Reviewing Editor has drafted this decision to help you prepare a revised submission.

Summary:

All reviewers agreed that the findings in this manuscript are novel and important. In has been known for some time that *T. brucei* is not restricted to the vasculature and can concentrate in various tissues. However, the experiments here show that a significant number of trypanosomes are present in skin and that these parasites can infect tsetse flies. These findings have important implications in terms of our understanding of the epidemiology of sleeping sickness and highlight the importance of determining the contribution that asymptomatic carriers may have in outbreak cycles.

Though overall the study is well-done there are some issues that must be addressed prior to our accepting the paper for publication. Major points are listed below but we would like to draw attention to two of the major points:

The quantification of trypanosomes based on luminescence should be given more prominence – do the tryps concentrate in the skin or are they more or less evenly distributed around the mouse?

Further, the data from human skin biopsies needs to be more convincing.

Essential revisions:

A recent publication by the Figueiredo group (Trindade S. et al., Cell Host Microbe) also showed that the skin of an infected mammal is a niche for *T. brucei* parasites. Although the Trindade paper was cited in the manuscript, it deserves more credit and discussion.

Figure 1 supported by [Supplementary-material SD7-data]. The experiment compares the trypanosome density in blood with a relative number in skin over a time course of infection. The authors conclude that the periodicity of infection is different. The clearest difference is on d24, it appears that this peak is due to one mouse d24I1, perhaps the authors should comment on this.

Also, it would help to have the blood trypanosome density for each timepoint (mouse) included in the supplementary data table.

Would it be better to plot blood parasitaemia against density in skin sections?

It is not clear how the leap is made from the bin-type scoring of low, moderate and large parasite numbers to the y-axis scale used? Can they further clarify – perhaps a different scale should be used?

Figure 2—figure supplement 1. The *T. gambiense* slide does not look like a skin section – it looks more like subcutaneous tissue – can the authors clarify? Is the enormous number of parasites seen in this section an outlier? How were the tissue sections for Figure 2 and Figure 2—figure supplement 1 chosen?

Figure 3 supported figure supplements and supplementary files

The results show that the localisation of the concentrations of trypanosomes in the skin varies over time in an individual mouse (3A). The detection of luminescence may correlate with density of trypanosomes in blood (how were parasitaemias below 5 x 104 / ml measured accurately?).

Are the following data available, if so it would help the reader to know:

Is the signal stronger if the trypanosomes are nearer the surface of the skin?

Would trypanosomes in the viscera be detected?

It would be important to know what fraction is the skin of the total mass of the mouse and what fraction of the bioluminescence is in the dissected skin? Do the trypanosomes concentrate in the skin or are they distributed around the mouse?

In panel 3B – Data from all mice should be shown in the graph- not just one representative mouse.

In Figure Supplement 3 it looks like different sections are shown with PAD1 and CRD staining but you state that you looked at both at the same time, i.e. co-staining. The figure legend should be more explicit and it would be better to show the same sections co-stained with PAD1 and CRD, if possible.

Figure 4 supported by figure supplements and supplementary files

The important data are shown in supplementary files and not in the main Figure 4. Thus, these data, [Supplementary-material SD10-data] and 5, should be brought into the main text, possibly as a table. It’s not clear exactly how these supplemental figures differ from one another – if there are important distinctions they should be made more clear, otherwise could these two supplemental figures can be combined?

Since the point is made that some flies are fed on bioluminescent areas and some are not, these data should also be included in [Supplementary-material SD10-data] and/or 5.

Lastly, the human data are not completely convincing. Some of the arrows (A and C) do not point to anything that looks different from all of the other purple cells. A negative biopsy sample as a control would be good. Are higher magnification images available?

Many figure panels and supplements are not fully discussed. Even in the legends there is frequently not sufficient detail for the reader to easily follow what was done. One example, some of the details included in the legend for Figure 2—figure supplement 1 would be helpful in the legend for Figure 1. The authors should go through the Results section and figure legends and make sure all information necessary for the reader to follow what was done in order to understand the results is there.

---

## [Author Response]

*Though overall the study is well-done there are some issues that must be addressed prior to our accepting the paper for publication. Major points are listed below but we would like to draw attention to two of the major points:*

*The quantification of trypanosomes based on luminescence should be given more prominence – do the tryps concentrate in the skin or are they more or less evenly distributed around the mouse?*

In the revised manuscript we have given the quantification and location of the trypanosomes in the skin more prominence creating a new table in the main text in place of [Supplementary-material SD10-data] and 5, expanding on the details of quantification and including an additional supplementary figure ([Supplementary-material SD5-data]) summarising the transmission experiment in an easily digestible format.

*Further, the data from human skin biopsies needs to be more convincing.*

To provide more convincing human biopsy data we prepared fresh slides from the onchocerciasis screening archive material held in DRC for staining with the trypanosome specific ISG65 antibody. Unfortunately, the samples were too fragile to survive the immunohistological procedure but we were able to *Giemsa* stain the slides. This allowed us to provide several convincing examples of trypanosomes in the skin of undiagnosed individuals at the higher magnification requested, including a transmission stumpy stage trypanosome in one individual. These are presented in Figure 5.

The reviewers also raised several essential revision points that we address in turn below:

*Essential revisions:*

*A recent publication by the Figueiredo group (Trindade S. et al., Cell Host Microbe) also showed that the skin of an infected mammal is a niche for T. brucei parasites. Although the Trindade paper was cited in the manuscript, it deserves more credit and discussion.*

We have revised the text to include further details of the conclusions of this publication but we stress that this paper did not show an association with trypanosome ‘adipose tissue forms’ and the skin as they were limited by the histological sections they examined.

*Figure 1 supported by [Supplementary-material SD7-data]. The experiment compares the trypanosome density in blood with a relative number in skin over a time course of infection. The authors conclude that the periodicity of infection is different. The clearest difference is on d24, it appears that this peak is due to one mouse d24I1, perhaps the authors should comment on this.*

We agree with the reviewers that the minor degree of periodicity observed in the time course is largely due to one mouse at d24 and added this detail to the text. We have adjusted our conclusions to state that there is little periodicity in the skin and trypanosomes, once present, remain in this extravascular compartment.

*Also, it would help to have the blood trypanosome density for each timepoint (mouse) included in the supplementary data table.*

The requested data has been included in [Supplementary-material SD2-data].

*Would it be better to plot blood parasitaemia against density in skin sections?*

*It is not clear how the leap is made from the bin-type scoring of low, moderate and large parasite numbers to the y-axis scale used? Can they further clarify – perhaps a different scale should be used?*

There was an error in the original submitted figure in that the blood parasitaemia scale was duplicated on both sides of the figure. This has now been adjusted to include a second scale with parasite skin density as an average of the semi-quantitative score over five high-powered fields.

*Figure 2—figure supplement 1. The T. gambiense slide does not look like a skin section – it looks more like subcutaneous tissue – can the authors clarify? Is the enormous number of parasites seen in this section an outlier? How were the tissue sections for Figure 2 and Figure 2—figure supplement 1 chosen?*

The reviewers are correct in stating that the presented image was not superficial skin but of the fascia directly beneath. This figure has now been adjusted to correctly show histological sections of the dermis. The high number of skin dwelling *T. b. gambiense* parasites relative to a typical *T. b. brucei* infection was observed in two of four mice and this is mentioned in the text and data presented.

*Figure 3 supported figure supplements and supplementary files*

The results show that the localisation of the concentrations of trypanosomes in the skin varies over time in an individual mouse (3A). The detection of luminescence may correlate with density of trypanosomes in blood (how were parasitaemias below 5 x 104 / ml measured accurately?).

The bioluminescence can originate from trypanosomes in the blood, but at low parasitaemia (<10^4^ parasites/ml) the bioluminescence from BSF is negligible and therefore hidden by the bioluminescence emitted by trypanosomes in the skin. Parasitaemia was estimated automatically with a cytometer (Muse, detection limit 5.10^2^ parasites/ml) and / or manually with a haemocytometer (Kova slides, detection limit 10^4^ parasites/ml) and these details have been highlighted in the Materials and methods.

*Are the following data available, if so it would help the reader to know:*

*Is the signal stronger if the trypanosomes are nearer the surface of the skin?*

*Would trypanosomes in the viscera be detected?*

*It would be important to know what fraction is the skin of the total mass of the mouse and what fraction of the bioluminescence is in the dissected skin? Do the trypanosomes concentrate in the skin or are they distributed around the mouse?*

The bioluminescence signal directly reflects the total number of living parasites in the entire organism, including blood and viscera, but the intensity of the signal decreases with tissue depth and light absorption varies according to the nature of the tissues. Therefore, at the end of all bioluminescence imaging experiments, mice were sacrificed and their organs dissected to individually check their bioluminescence: in addition to the skin, only the spleen, some lymph nodes and adipose tissues were seen positive in several mice, suggesting that viscera were not privileged sites for parasite development in our conditions. The bioluminescent profile of mouse 3A dissected organs was added as Figure 3—figure supplement 1. In total, the bioluminescence detected in this experiment is likely to originate mostly, yet not only, from parasites in the skin. Estimating the total mass of the skin is not easy and may not be so informative as skin parasite density varies greatly in time and space over the course of infection. Moreover, because bioluminescence decreases with the depth of the source, it is not directly cumulative: i.e. the sum of bioluminescence intensities from dissected organs does not correspond to that of the entire living organism.

*In panel 3B – Data from all mice should be shown in the graph- not just one representative mouse.*

This figure has been revised to include data from multiple experiments rather than one representative mouse.

*In Figure Supplement 3 it looks like different sections are shown with PAD1 and CRD staining but you state that you looked at both at the same time, i.e. co-staining. The figure legend should be more explicit and it would be better to show the same sections co-stained with PAD1 and CRD, if possible.*

Co-staining was not possible because the two available antibodies are polyclonal rabbit antibodies and can therefore not be used at the same time. We have clarified this in the Materials and methods section.

*Figure 4 supported by figure supplements and supplementary files*

*The important data are shown in supplementary files and not in the main Figure 4. Thus, these data, [Supplementary-material SD10-data] and 5, should be brought into the main text, possibly as a table. It’s not clear exactly how these supplemental figures differ from one another – if there are important distinctions they should be made more clear, otherwise could these two supplemental figures can be combined?*

*Since the point is made that some flies are fed on bioluminescent areas and some are not, these data should also be included in [Supplementary-material SD10-data] and/or 5.*

All data were re-organised and are now more clearly presented and annotated. A new table was created from [Supplementary-material SD10-data] and 5 and is in the main text. This table presents all the xenodiagnoses performed after 48h. For differential xenodiagnosis, flies were indeed fed on skin regions with various levels of bioluminescence that were further used to estimate the number of parasites per skin surface unit ranging from 0 to 6.7x10^7^ parasites / cm^2^ and presented in Table 1. A diagram illustrating this point has been added ([Supplementary-material SD5-data]).

*Lastly, the human data are not completely convincing. Some of the arrows (A and C) do not point to anything that looks different from all of the other purple cells. A negative biopsy sample as a control would be good. Are higher magnification images available?*

*Many figure panels and supplements are not fully discussed. Even in the legends there is frequently not sufficient detail for the reader to easily follow what was done. One example, some of the details included in the legend for Figure 2—figure supplement 1 would be helpful in the legend for Figure 1. The authors should go through the Results section and figure legends and make sure all information necessary for the reader to follow what was done in order to understand the results is there.*

Figure legends have been expanded and clarified.